SciPost Physics

Submission

# Dynamical Generation of Spin Current and Phase Slip in Exciton-Polariton Condensates

S. J. Chen[1], S. D. Guo[1], Bo Xiong[1,2*]

**1** Department of Physics, Nanchang University, 330031 Nanchang, China
**2** Skolkovo Institute of Science and Technology, Novaya Street 100, Skolkovo 143025, Russian Federation
* stevenxiongbo@gmail.com

April 9, 2021

## Abstract

We show that how to generate propagation of spin degree in spin-symmetric exciton-polariton condensates in a semiconductor microcavity. Due to the stimulated spin-dependent scattering between hot excitons and condensates, exciton polaritons form a circular polarized condensate with spontaneous breaking of the spin rotation symmetry. The spin antiferromagnetic state is developed evidently from the density and spin flow pumped by localized laser source. The low energy spin current is identified where the steady state is characterized by the oscillating spin pattern. Finally, we predict via simulation how to dynamical generation of phase slip where ring-shape phase jump shows the behavior of splitting and joining together.

# 1 Introduction

Recently, in semiconductor microcavities with quantum wells sandwiched between highly reflective mirrors, the strong coupling is achieved between excitons and photons [1–4]. Such coherent light-matter particles called exciton-polaritons obey the Bose-Einstein statistics and thus condense at critical temperatures ranging from tens Kelvin [5–7] till several hundreds Kelvin [8, 9], which exceeds by many orders of magnitude the Bose-Einstein condensation temperature in atomic gases. Recently, electrically pumped polariton laser or condensation was realized based on a microcavity containing multiple quantum wells [10,11]. Considering the high transition temperatures and high tunability from pumping source, semiconductor microcavities are perfectly suited for studies of macroscopically collective phenomenon and have initiated the fascinating research on the polariton quantum hydrodynamics.

The polaritons have two allowed spin projections on the structure growth axis, $\pm 1$, corresponding to right- and left- circular polarizations of photons. In diverse semiconductor materials like GaAs/GaAlAs [12], Si [13], organic single-crystal microcavity $SiN_x/SiO_2$ [14] and so on, spin injection and detection has been successfully realized which is one of the key ingredients for functional spintronics devices. A number of prominent spin-related phenomena both in interacting and in noninteracting polariton systems have already been predicted and observed in the microcavities, such as, spontaneous polarization [15–21], polarization multistability [22–27], optical spin Hall effect [28–34] and topological insulator [35–40], spin Zeeman and Meissner effect [41–43].

Spin degrees of freedom in two-dimensional exciton-polaritons superfluid can drastically change elementary topological vortices referred to as half-quantum vortices (HQV) [44–48] which are characterized by a half-integer value of vorticity in contrast to the regular quantum vortex [49–56] where the vorticity takes only integer values. Usually HQV carry only one half-integer topological charge originating both from the superfluid current proportional to $\nabla\theta$, and from $\pi$ spin disclinations superimposed as a result of Berry's phases induced by spin rotations [57]. Relevant ideals of half vortices have been discussed in A phase of $^3$He [58–60], in triplet superconductors $Sr_2RuO_4$ [61] and spinor atomic Bose-Einstein condensates [62–65] with two different spin components where HQV is just residing in one of components [66–70].

However, precise coherent control of spin polarization, propagation and topological defects in exciton-polariton condensates still remains a core challenge. Here, we address this problem, and demonstrate exciton-polariton condensates will not only show spontaneous polarization and also coherent propagation of the pseudospin under nonlocal spin injection. When taking into account incoherent hot exciton reservoir scattered into coherent states, dramatically enhanced spin-polarized signal can be observed at the appropriate pumping regime. Moreover, the coherent spin antiferromagnetic state can also be identified and manipulated by spin-symmetric pumping source. Additionally, cavity engineering allows us to the dynamic generation of phase slip where ring-shape phase jump shows the behavior of splitting and joining together induced by incoherent reservoir as a result of effective gauge field.

# 2 Physical Background.

In the absence of external magnetic field the "spin-up" and "spin-down" states $\sigma = \pm$ of noninteracting polaritons, or their linearly polarized superpositions, are degenerate corresponding to the right ($\sigma_+$) and left ($\sigma_-$) circular polarizations of external photons. The

spinor nature of exciton polaritons can therefore be manifested since the spin are essentially free in semiconductor microcavities. To illustrate the fully degenerate spinor nature, and as a first step, the Zeeman energy must be much smaller than the interaction energy. Thus we shall consider only the case of zero magnetic field achieving a good approximation in the following. Since the interaction between exciton polaritons depends on their total spins (singlet or triplet), their spin states may be changed after the scattering. The spin-dependent interactions cause the polariton spin states exchange. Moreover, additional mixing may comes from the longitudinal-transverse (LT) splitting of polaritons (referred to as the Maialle mechanism) [71] and from structural anisotropies [72].

The low energy dynamics is therefore described by a pairwise interaction that is spin-rotation invariant and preserves the spin of the individual exciton polaritons. The general form of this interaction is $\hat{V}(\mathbf{r}_1 - \mathbf{r}_2) = \delta(\mathbf{r}_1 - \mathbf{r}_2)\sum_{F=0}^{2f} g_F \cdot \hat{P}_F$ where $g_F = 4\pi\hbar^2 a_F/M$, $M$ is the mass of exciton polaritons, $\hat{P}_F$ is the projection operator which projects the pair 1 and 2 into a total spin F state, and $a_F$ is the s-wave scattering length in the total spin F channel. For exciton polaritons of $f = 1$ bosons, interaction has form $\hat{V} = g_0 \cdot \hat{P}_0 + g_2 \cdot \hat{P}_2$. In terms of nonlinear optics, the coupling coefficients of polarization independent $c_0$ and so-called linear-circular dichroism $c_2$ can be estimated through the matrix elements of the polariton-polariton scattering in the singlet and triplet configurations.

It is convenient to write the Bose condensate $\Psi_a(\mathbf{r}) \equiv <\hat{\psi}_a(\mathbf{r})>$ as $\Psi_a(\mathbf{r}) = \sqrt{n(\mathbf{r})}\zeta_a(\mathbf{r})$, where $n(\mathbf{r})$ is the density, and $\zeta_a$ is a normalized spinor $\zeta^+ \cdot \zeta = 1$. It is obvious that all spinors related to each other by gauge transformation $e^{i\theta}$ and spin rotations $\mathcal{U}(\alpha, \beta, \gamma) = e^{-iS_x\alpha}e^{-iS_y\beta}e^{-iS_z\gamma}$ are degenerate, where $(\alpha, \beta, \gamma)$ are the Euler angles.

The non-equilibrium dynamics of polariton condensates is described by a Gross-Pitaevskii type equation for the coherent polariton field, which should be coupled to a hot-excitons reservoir excited by the nonresonant exciting pump. The model is, however, generalized to take into account the polarization degree of freedom of hot exciton. In this approach, instead of polarization independent scattering, we must take into account dichroism scattering between hot exciton and coherent polariton field.

Let us turn to the pseudospin representation, then the local spin density $\vec{s}$ at the position $\mathbf{r}$ and time $t$ is $\vec{s}(\mathbf{r}, t) = \Psi^\dagger(\mathbf{r}, \mathbf{t})\hat{\vec{s}}\Psi(\mathbf{r}, \mathbf{t})$, where $\hat{\vec{s}} = (\hbar/2)\hat{\vec{\sigma}}$ with $\hat{\vec{\sigma}}$ being the Pauli matrices. The usual definition of the free-particle probability current $\mathbf{J}_n = \mathrm{Re}\left[\Psi^\dagger(\mathbf{r}, \mathbf{t})\frac{\hat{\mathbf{P}}\hat{I}}{m}\Psi(\mathbf{r}, \mathbf{t})\right]$, where $\hat{I}$ is the identity, and probability spin current $\mathbf{J}_{\vec{s}} = \mathrm{Re}\left[\Psi^\dagger(\mathbf{r}, \mathbf{t})\frac{\hat{\mathbf{P}}\vec{s}}{m}\Psi(\mathbf{r}, \mathbf{t})\right]$. In addition, the emergent magnetic monopoles defined by analogy with Maxwell's equation as $\nabla \cdot \vec{s}$ can be realized and characterized by a divergent in-plane pseudospin pattern, that have been present in magnetically frustrated materials, spin-ice [73–79], magnetic nanowires [80] and atomic spinor Bose-Einstein condensates [81, 82]. The dynamics of each spin under the effect of magnetic field is governed by the precession equation $\partial_t\mathbf{S} = \mathbf{H} \times \mathbf{S}/\hbar$. The total effective magnetic field $\mathbf{H}$ represents the sum of the field responsible provided by the spin dependent and independent polariton-polariton interactions and polariton-hot exciton interaction (LT splitting $\mathbf{H}_{LT}$ is assumed to be negligible in high density regime). Very different from those isolated or closed system, the dynamic of spin pattern in such open-dissipative system is crucially determined by the pump source. We will go into further details in the following.

# 3 Theoretical Model.

In the following, we study the propagation of polarized polariton in the a planar microcavity and generation of spin polarization, spin current and the observability of the HQV,

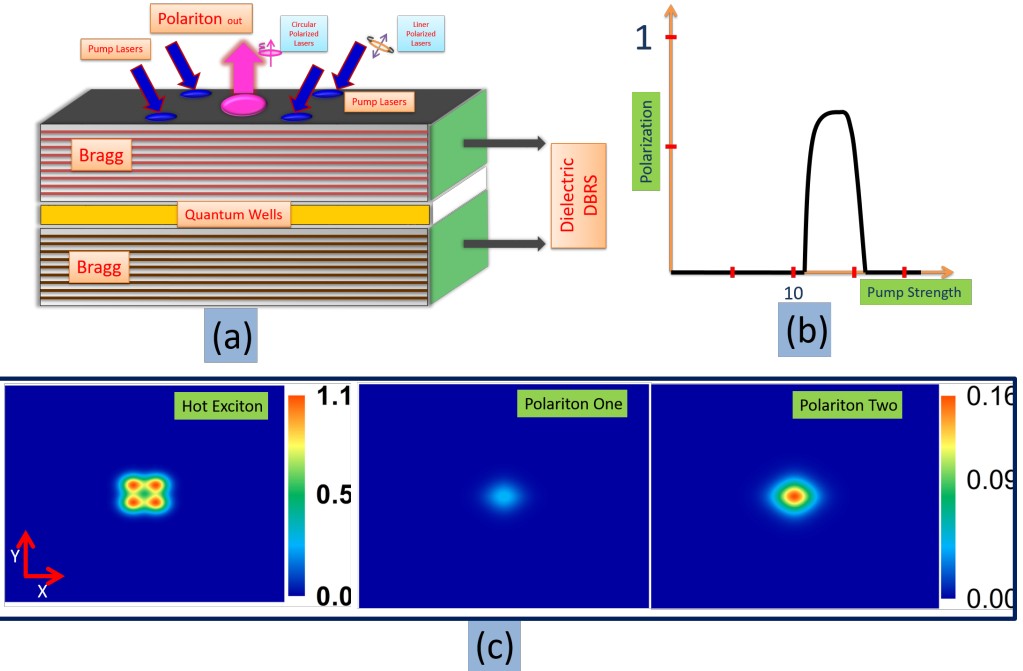

Figure 1: (Color online) The spontaneously circular polarization of spinor condensate non-resonant pumped by linearly polarized laser. (a) Proposed scheme to experimentally stimulating spontaneous circular polarization by nonpolarized laser beam. (b) Spinor is polarized when the laser power is larger than first threshold value, however, unpolarized after laser power is above second threshold value. (c) Density distribution of hot exciton (left picture which has the same profile for both components) and spinor polariton (middle and right pictures for each components) in real space. Here, simulations are in the absence of disorder for 4 pumping points with a small radius 1.54 $\mu$m. The size of profile is 24x24 and the other parameters used in the simulations are shown in the paper.

in realistic structures. The equation of motion for the spinor polariton wave function reads [83–86]

$$i\hbar\partial_t\psi_\pm(\mathbf{r}) = \left\{ -\frac{\hbar^2}{2m}\nabla^2 + \frac{i\hbar}{2}\left(g_2 n_{R\pm} + h_2 n_{R\mp} + \beta_2|\psi_\pm|^2 + f_2|\psi_\mp|^2 - \gamma_C\right) + V_{ext}(\mathbf{r})\right\}\psi_\pm(\mathbf{r})$$
$$+ \left\{\hbar\left(\beta_1|\psi_\pm|^2 + f_1|\psi_\mp|^2\right) + V_R(\mathbf{r})\right\}\psi_\pm(\mathbf{r}), \tag{1}$$

where $\psi_\sigma$ represents the condensed field, with $\sigma = \pm$ representing the spin state of polaritons with effective mass $m$. $\gamma_C$ represents the coherent polariton decay rate. $\beta_1$ and $f_1$ is the spin-conserved and spin-exchange polariton-polariton interaction strength, respectively. $n_{R\sigma}$ is the density of the incoherent hot exciton reservoir. And here, $V_R(\mathbf{r}) = \hbar\left[g_1 n_{R\pm} + h_1 n_{R\mp} + \Omega P_\pm(\mathbf{r})\right]$ represents spin-conserved and spin-exchange interactions with hot exciton reservoir where $P_\pm(\mathbf{r})$ is the spatially dependent pumping rate and $g_1, h_1, \Omega > 0$ are phenomenological coefficients to be determined experimentally. $V_{ext}(\mathbf{r})$ represents the static disorder potential in semiconductor microcavities, which is typically chosen as the same for both component polaritons. $g_2 n_{R\pm}$ and $h_2 n_{R\mp}$ are related with the condensation rate in that growth of condensate are stimulated by hot excitons with same spin or cross spin, respectively [87]. $\beta_2$ and $f_2$ are the same-spin and cross-spin nonradiative loss rates, respectively.

The equation 1 of condensate is coupled to a rate equation describing the time evolution of density $n_{R\sigma}$ of incoherent hot exciton as:

$$\partial_t n_{R\pm} = -\Gamma n_{R\pm} - \left[g_2|\psi_\pm|^2 + h_2|\psi_\mp|^2\right]n_{R\pm} + P_\pm, \tag{2}$$

where the reservoir relaxation rate $\Gamma$ is much faster than that of condensate $\Gamma \gg \gamma_C$ where the Gaussian pump laser $P_\pm = W$ is assumed nonpolarized (corresponding to linear or horizontal polarization) providing a sufficient large occupation in momentum space of incoherent hot exciton. The stimulated emission of the hot exciton reservoir into condensate is taken into account by the term $\left[g_2|\psi_\pm|^2 + h_2|\psi_\mp|^2\right]n_{R\pm}$. The spatial diffusion rate of reservoir density has been neglected. In the following, we solve the coupled Eqs. 1 and 2 numerically starting from a small random initial condition. As we can see that, the time evolution of the system has been obtained until a steady state is reached independent of the initial noise.

# 4 Steady State.

## 4.1 Spatially homogeneous system

Let us begin with some analytical consideration on spinor condensate. In the homogeneous case, i.e., under a spatially homogeneous pumping and in the absence of any external potential, Eqs. 1 and 2 admit analytical stationary spinor configuration. Below the pumping threshold, the condensate remains unpopulated, while the reservoir grows linearly with the pump intensity as $n_{R\pm} = W/\Gamma$. At the threshold pump intensity $W^{th}$, the stimulated emission rate exactly compensates the losses $g_2 n_{R\pm} + h_2 n_{R\mp} = \gamma_C$ and condensate becomes populated dynamically. We notice that threshold pump intensity becomes $W^{th} = \Gamma\gamma_C/(g_2 + h_2)$. Above the threshold, the reservoir density is homogeneous $n_{R\pm} = W/\left(\Gamma + g_2|\psi_\pm|^2 + h_2|\psi_\mp|^2\right)$, from this, we obtain

$$Z_R \sim -\frac{W(g_2 - h_2)}{\Gamma^2 + \Gamma(g_2 + h_2)n_c}Z_C, \tag{3}$$

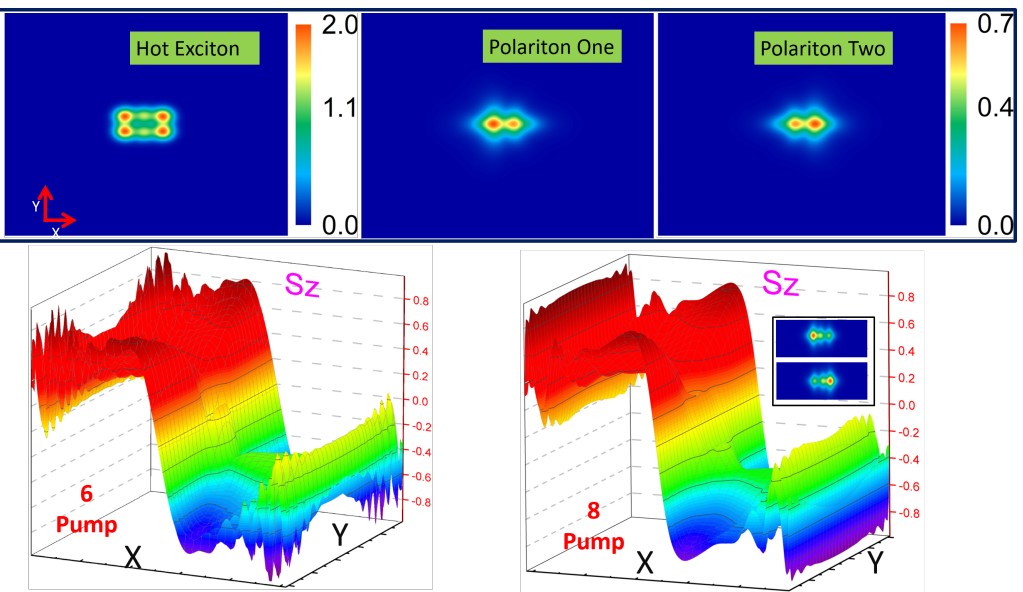

Figure 2: (Color online) The spontaneously circular polarization of spinor condensate non-resonant pumped by 6 and 8 linearly polarized laser, respectively. Top panel: density distribution of hot exciton (left picture which has same profile for both components) and spinor polariton (middle and right pictures for each components) in real space for 6 pumping points. Bottom panel: distribution of magnetic polarization along the Z axis for 6 pumping points (left picture) and 8 pumping points (right picture, where inset shows density distribution of two component polariton). The size of profile is 24x24 and the other parameters used in the simulations are the same as those in the Fig. 1.

here, we have defined reservoir polarization $Z_R = n_{R+} - n_{R-}$, condensate polarization $Z_C = |\psi_+|^2 - |\psi_-|^2$ and condensate total density $n_c = |\psi_+|^2 + |\psi_-|^2$. As long as $g_2 \neq h_2$, condensate polarization is directly proportional to the reservoir polarization.

From the Eqs. 1, we find that the condensate density is

$$n_c \sim \frac{\left(W - W^{th}\right)}{\gamma_C} \cdot \frac{1}{1 - \frac{1}{2}\left(\frac{W}{W^{th}} + \frac{\beta_2 + f_2}{g_2 + h_2}\frac{\Gamma}{\gamma_C}\right)}, \tag{4}$$

and condensate polarization satisfy

$$M_C Z_C = 0. \tag{5}$$

where

$$M_C = \left(4W g_2 h_2 + \Gamma^2 \left(\beta_2 - f_2\right) - \frac{W \Gamma^2 \gamma_C^2}{W^{th} \cdot W^{th}}\right).$$

Except very stringent condition $M_C = 0$, otherwise, magnetization of condensate is always zero, i.e., $Z_C = 0$ seen from the Eq. 5. If assuming cross-spin radiative and nonradiative loss rates is negligible, magnetization condition $M_C = 0$ leads to following condition for pump laser power

$$W = \frac{\gamma_C^2}{\beta_2 \left(W^{th}\right)^2} = \frac{g_2^2}{\beta_2 \Gamma^2}, \tag{6}$$

therefore, considering necessary condition $W > W^{th}$, we find following conditon should be satisfied for spontaneous magnetization of condensate,

$$\frac{g_2^3}{\beta_2 \gamma_C \Gamma^3} > 1.$$

If assuming condensate wave function takes the form $\psi_\pm\left(\mathbf{r}\right) = \sum \psi_{\mathbf{k}_\pm \omega_\pm} e^{i(\mathbf{k}_\pm \cdot \mathbf{r} - \omega_\pm t)} \sim \psi_{\mathbf{0}\pm} e^{i(\mathbf{k}_\pm \cdot \mathbf{r} - \omega_\pm t)}$, we find spectrum as

$$\omega_\pm = \frac{\hbar k_\pm^2}{2m} + \tilde{\Omega}_\pm W, \tag{7}$$

where

$$\begin{aligned}\tilde{\Omega}_\pm = \Omega &+ \frac{\left(\beta_1 + f_1\right) n_c \pm \left(\beta_1 - f_1\right) Z_C}{2W} \\ &+ \frac{2\left(g_1 + h_1\right)\Gamma + G \cdot n_c \pm H \cdot Z_C}{2\left[\Gamma^2 + \Gamma\left(g_2 + h_2\right) n_c + A\right]},\end{aligned}$$

here, wave vector $\mathbf{k}_\pm$ and frequency $\omega_\pm$ remains so far undetermined, and coefficient $G = g_1 g_2 + g_1 h_2 + g_2 h_1 + h_1 h_2$, $H = \left(g_1 h_2 + g_2 h_1 - g_1 g_2 - h_1 h_2\right)$. However, from Eq. 7, we find frequency difference between two component is given by

$$\omega_+ - \omega_- = \frac{\hbar\left(k_+^2 - k_-^2\right)}{2m} + \Delta\tilde{\Omega}, \tag{8}$$

here,

$$\begin{aligned}\Delta\tilde{\Omega} \sim Z_C \{&\left(\beta_1 - f_1\right)/W \\ &- \left(g_1 - h_1\right)\left(g_2 - h_2\right)/\left[\Gamma^2 + \Gamma\left(g_2 + h_2\right)n_c + A\right]\},\end{aligned}$$

where $A$ is high order term of density and polarization $A = \left(g_2 + h_2\right)^2 \left(n_c^2 + Z_C^2\right)/4$ which can be dominant term for the large density and polarization. Interestingly, we can see that energy gap is polarization dependence. In particular, when $\beta_1 \simeq f_1$ or large enough laser power $W$, polarization dependence of frequency difference disappears.

## 4.2 Local density and spin approximation

In the presence of an inhomogeneous laser pump $W(\mathbf{r})$ (or multiple pump $W_i(\mathbf{r})$), much richer phenomena will be represented, such as, spin domain formation, emergent magnetic monopole, generation of half vortex and so on. Under inhomogeneous laser pump, we thus look for stationary spinor polariton wave function as following form

$$\Psi = \begin{pmatrix} \psi_+ \\ \psi_- \end{pmatrix} = \sqrt{\rho(\mathbf{r})}\zeta(\mathbf{r})e^{-i(\phi(\mathbf{r})-\omega_\pm t)}, \tag{9}$$

where $\rho(\mathbf{r})$ and $\phi(\mathbf{r})$ are the local density and phase of the condensate, and $\zeta(\mathbf{r})$ is spinor function. We are going to assume that the local pump imposes a boundary condition for the spinor function at each pumping spot $r_p$: $\lim_{r\to r_p}\zeta(\mathbf{r}) = \lambda$, $\lim_{r\to r_p}\mathbf{k}_C(\mathbf{r}) = 0$, here, we have defined local condensate density wave vector $\mathbf{k}_C(\mathbf{r}) = \nabla_\mathbf{r}\phi(\mathbf{r})$. In the following, the dimensionless form of the model can be obtained by using the scaling units of time, energy, and length as: $T = 1/\gamma_C$, $E = \hbar\gamma_C$, $L = \sqrt{\hbar/m\gamma_C}$, respectively.

Inserting Eq. 9 into the Eqs. of motion 1 and 2, one obtains the following set of conditions for stationary solution:

$$\omega_\pm = -\frac{1}{2}\left(\frac{\nabla^2\sqrt{\rho}}{\sqrt{\rho}} + \frac{\nabla^2\zeta_\pm}{\zeta_\pm} + 2\frac{\nabla\sqrt{\rho}\cdot\nabla\zeta_\pm}{\sqrt{\rho}\zeta_\pm} - k_C^2\right)$$
$$+\frac{1}{\gamma_C}\left(\beta_1|\zeta_\pm|^2\rho + f_1|\zeta_\mp|^2\rho + g_1 n_{R\pm} + h_1 n_{R\mp}\right) + \frac{\Omega W}{\gamma_C}, \tag{10}$$

and

$$\frac{1}{2}\left(g_2 n_{R\pm} + h_2 n_{R\mp} + \beta_2|\zeta_\pm|^2\rho + f_2|\zeta_\mp|^2\rho - \gamma_C\right)$$
$$+\frac{1}{2}\nabla\cdot\mathbf{k}_C(\mathbf{r}) + \frac{\nabla\sqrt{\rho}\cdot\mathbf{k}_C(\mathbf{r})}{\sqrt{\rho}} + \frac{\mathbf{k}_C(\mathbf{r})\cdot\nabla\zeta_\pm}{\zeta_\pm} = 0, \tag{11}$$

and

$$\Gamma n_{R\pm} + \left(g_2|\zeta_\pm(\mathbf{r})|^2 + h_2|\zeta_\mp(\mathbf{r})|^2\right)\rho(\mathbf{r})n_{R\pm} = W(\mathbf{r}). \tag{12}$$

Different from the single component condensate, now in Eq. 10, the quantum pressure terms are not only originated from density $\nabla^2\sqrt{\rho}$ but also from the spinor $\nabla^2\zeta$ and even spin-density coupling $\nabla\sqrt{\rho}\cdot\nabla\zeta$. Moreover, in Eq. 11, besides the current divergence term, we can see the more terms appeared which is originated from coupling of superfluid current with density pressure $\nabla\sqrt{\rho}\cdot\mathbf{k}_C(\mathbf{r})$ or spin pressure $\mathbf{k}_C(\mathbf{r})\cdot\nabla\zeta$.

We can make local density approximation (LDA) and local spin approximation (LSA) if the spatial variation of the laser pump $W(\mathbf{r})$ is smooth enough. In such approximations, the quantum pressure term in Eq. 10 and 11 can be neglected. Interestingly, similar to the homogeneous case, the condensate density profile and polarization is still given by the same Eq. 4 and Eq. 5, respectively, except homogeneous laser pump $W$ is replaced with local value $W(\mathbf{r})$ in there.

Under the Gaussian laser pump profile, we can look for cylindrically symmetric stationary solutions. The condensate frequency $\omega_\pm$ is

$$\omega_\pm = \frac{\tilde{\Omega}_\pm \cdot W}{\gamma_C}, \tag{13}$$

which is determined by the boundary condition that the local density wave vector vanishes

$\mathbf{k}_C\left(\mathbf{r}=\mathbf{r}_p\right)=0$ at the center of the each pumping spot. Here,

$$\tilde{\Omega}_\pm = \Omega + \frac{\left(\beta_1+f_1\right)\rho \pm \left(\beta_1-f_1\right)\rho S_Z}{2W} + \frac{2\left(g_1+h_1\right)\Gamma + \left[G\cdot\rho \pm H\cdot\rho S_Z\right]}{2\left[\Gamma^2+\Gamma\left(g_2+h_2\right)\rho + B\cdot\rho^2\right]}, \tag{14}$$

from here, we can find frequency difference between two component as

$$\omega_+ - \omega_- = \frac{\Delta\tilde{\Omega}\cdot W}{\gamma_C}, \tag{15}$$

here,

$$\begin{aligned}\Delta\tilde{\Omega} &= \tilde{\Omega}_+ - \tilde{\Omega}_- \\ &= \rho\left(\mathbf{r}_p\right)S_Z\left(\mathbf{r}_p\right)\left\{\left(\beta_1-f_1\right)/W\right. \\ &\quad \left. - \left(g_1-h_1\right)\left(g_2-h_2\right)/\left[\Gamma^2+\Gamma\left(g_2+h_2\right)\rho + A\rho^2\right]\right\},\end{aligned}$$

here, we have defined condensate polarization $S_Z\left(\mathbf{r}_p\right)=\left|\zeta_+\left(\mathbf{r}_p\right)\right|^2 - \left|\zeta_-\left(\mathbf{r}_p\right)\right|^2$ and co-efficient of density square term $B=\left(g_2+h_2\right)^2\left(1+S_Z^2\right)/4$, which has maximal value $\left(g_2+h_2\right)^2/2$ for the total polarization $\pm 1$. Interestingly, we can see that energy gap is polarization dependence. In particular, when $\beta_1 \simeq f_1$ or large enough laser power, polarization dependence of frequency difference disappears.

Local density wave vector $\mathbf{k}_C\left(\mathbf{r}\right)$ of condensate is reaching maximal value with the condensate density decreased and spin polarized away from the pumping center. Polaritons condense at the laser spot position has a large blueshifted energy due to their interactions with uncondensed hot excitons, thus within a short time, these interaction energy will lead to the motion of polariton initially localized at pumping point. In particular, spontaneous polarization may happen because polarization may lower the frequency obviously under the laser power is large enough as we can see from Eq. 14. Therefore, spin domain, spin current and topological defect may be formed under such appropriate condition.

In the following, through extensive numerical simulations of the Eq. 1 coupled to the reservoir evolution Eq. 2, above analytical results have been approved, such as, the dynamical formation of spin domain, spin current and half vortex for a wide range of pump parameters obviously available within state-of-the-art techniques.

## 5 Numerical Results for Spontaneous Polarization.

Eqs. of motion 1 and 2 can be solved numerically with the initial condition $n_{R\sigma}(x,y,t)\approx 0$, $\psi_\sigma(x,y,t)\approx 0$. The parameters of the pump are chosen according to the related experiments [32–34, 48] which study the optical spin hall effect, tunable spin textures and half solitons. In our calculations the following parameters are used typically for state-of-the-art GaAs-based microcavities: the polariton mass is set to $m=10^{-4}\ m_{\mathrm{e}}$ where $m_{\mathrm{e}}$ is the free electron mass; the decay rates are chosen as $\gamma_C=0.152\ \mathrm{ps}^{-1}$ and $\Gamma=3.0\gamma_C$; thus, the scaling units of time, energy and length are 6.58 ps, 0.1 meV, and 1.54 $\mu$m, respectively; the interaction strengths are set to $\hbar\beta_1=40\ \mu\mathrm{eV}\ \mu\mathrm{m}^2$, $f_1=-0.1\beta_1$, $g_1=2\beta_1$, $h_1=-0.2\beta_1$; the condensation rate are set to $\hbar g_2=0.16\ \mathrm{meV}\ \mu\mathrm{m}^2$, $\hbar h_2=0.016\ \mathrm{meV}\ \mu\mathrm{m}^2$, and condensation loss rate $-\hbar\beta_2=0.16\ \mathrm{meV}\ \mu\mathrm{m}^2$, $\hbar f_2=0.016\ \mathrm{meV}\ \mu\mathrm{m}^2$. In our simulation, the dimensionless scattering coefficient for each interaction term has been tuned carefully in order to get the physical phenomena we want due to complicated nonlinear effects. From an experimental point of view, the dimensionless

interaction parameters must be adjusted to match pump intensity. The pump intensity was chosen according to the experimentally measured blueshift of the polariton condensate, and its profile is Gaussian shape as:

$$W(\mathbf{r}) = \frac{w_0}{\pi w_1^2} \sum_{i=1}^{n} e^{\frac{-(x-x_i)^2 - (y-y_i)^2}{w_1^2}},$$

here, for a typical case, $w_1 = 1.0$, $|x_i| = |y_i| = 1.5$, and $w_0$ is tuned accordingly.

As expected, Eqs. of motion 1 and 2 tend to settle to a steady state with a spontaneously circular polarization under increasing laser power as shown in Fig. 1(b). Threshold laser power for spontaneously circular polarization is greater than that of starting condensation which can be understood from our derived Eqs. 5 and 6. The coherent polarized polaritons ballistically fly away from the laser spot due to their interactions converted into kinetic energy of coherent polariton. In particular, the circular polarization rapidly saturates with increasing the pumping power and may lead to an almost full polarization [17–19]. Surprisingly, full circular polarization state will change immediately back to the linear polarization with further increasing the laser power (i.e., the density of condensate exceeding a threshold value). Such phenomenon can be understood from Eqs. 10 and 11 where various quantum pressure terms will take important roles. Moreover, as shown in the Fig. 1(c), density profiles of incoherent hot exciton and polariton condensate represent linear and circular polarization, respectively within spontaneously circular polarization regime. As we can see that, while unpolarized hot excitons experience a limited diffusion, polarized polaritons ballistically fly away from the laser spot due to the conversion between interactions energy and kinetic energy.

Fig. 2 show the density distribution of incoherent hot exciton and coherent polariton condensate under six and eight unpolarized pumping laser points. Interestingly, the neighbouring condensed polaritons are polarized with opposite polarization as can be seen from $s_z$ distribution clearly. Moreover, steady state with magnetic domain wall formation has been obtained and characterized by vanishing total magnetization. Such phenomena is fundamentally related with emergent effective magnetic field by the inhomogeneous pump laser as we mentioned before. Furthermore, other interesting magnetic textures may be formed from the evolution of Eqs. of motion 1 and 2. In the following, we will address the question how to generate the density current, spin current, phase fluctuation and slip via tuning pumping (or geometrical) source.

# 6   Density Current, Spin Current, Phase Slip.

Physically, condensed fluid is a long-range cooperative phenomenon characterized by long-range correlation and coherent ordering of the momenta of particle. The various correlation function may imply net surface currents and orbital angular momentum appearing in this system. Therefore, It is important to study the density and spin current, and furthermore, study how to generate and control them. In the following, we will address these questions by suddenly shifting pumping laser position by a distance. Interestingly, we find that a steady current can be generated apparently. In particular, if shifted the pumping laser is linear or circular polarized, we observe large phase fluctuation where ring-shape phase jump shows the behavior of splitting and joining together. The above-mentioned behaviors may be understood from emergent effective gauge field caused by externally pumped incoherent reservoirs.

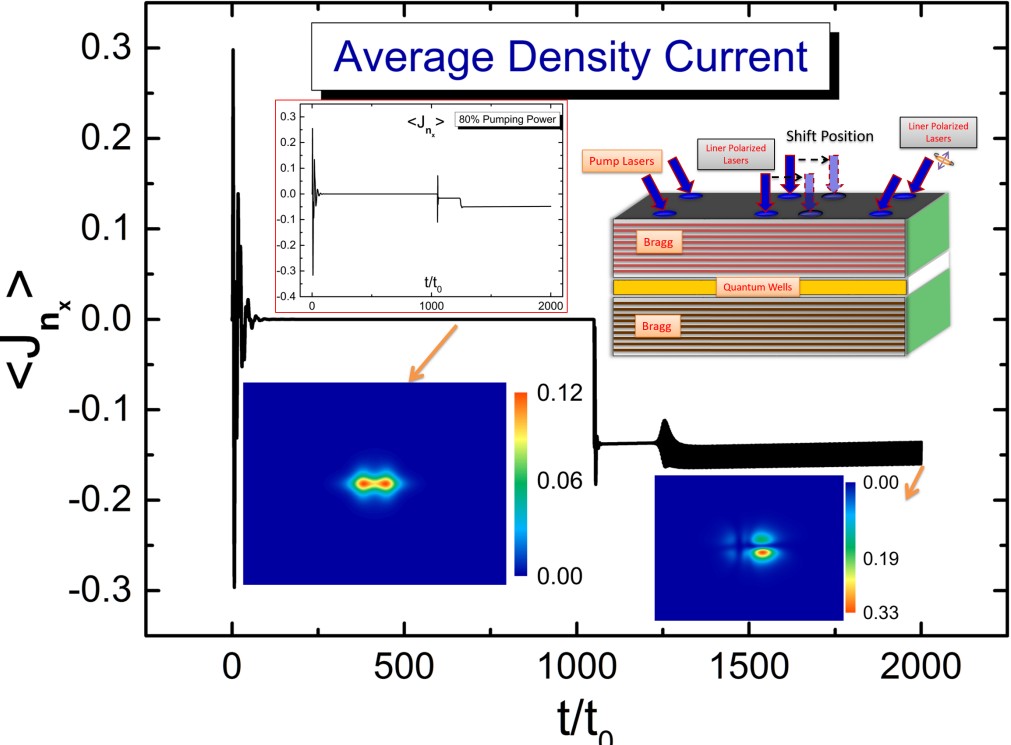

Figure 3: (Color online) Normalized average density current $J_{n_x}$ of condensate which is non-resonantly excited by 6 points laser with linear polarization. The insets shows the total density profile before and after shifting position of two middle lasers (see the schematic picture) along the x direction, and also shows $J_{n_x}$ under decreasing the pumping power to 80%. The size of profile is 24x24 and the other parameters used in the simulations are the same as those in the Fig. 1.

## 6.1 Density current

First, we numerically simulate time evolution of Eqs. of motion 1 and 2 under pumped by six linearly polarized laser and then, suddenly shifting two middle laser's position. The results are shown in the Fig. 3 for the average density current, which are normalized by the total density of condensate. As is shown in the Fig. 3, a large-amplitude oscillation appears within a short time when switching on the pumping lasers. With time evolution, oscillation decays very quickly and disappear at 60 unit of time. The appearance of such oscillation can be understood from the large overlap of incoherent hot exciton and coherent polariton which leads to the large repulsive force in the beginning. Then, with coherent polariton's diffusion under such repulsive force, condensate stay in a steady state with zero averaged current, which means a balanced configuration in momentum space of condensate.

Second, we want to generate steady current without decay by breaking above balanced configuration. Therefore, we suddenly shift two middle laser's position at time 1050 (referring to the schematic picture in the inset of Fig. 3). Interestingly, a persistent current with small oscillation can be observed clearly and it's amplitude is centred at -0.15. The appearance of such persistent current can be understood from breaking balanced-momentum configuration due to changing interaction energy between different part of condensate. Moreover, accompanied fast small-amplitude oscillation can be understood as surface oscillation modes which are moved back and forth due to confinement by the pumping laser. Furthermore, such oscillation can be suppressed by lowering the pumping power completely as is shown in the inset of Fig. 3, where pumping power drops up to 80 percent of previous case. However, we can not generate persistent spin current by using above method. Therefore, we will address this issue in the following section.

## 6.2 Spin current

Polariton condensates are excellent candidates for designing novel spin-based devices at room temperature due to their many features, such as strong optical nonlinear response, spin polarization properties, and fast spin dynamics. Therefore, in the following, we will show how to generate spin transportation of coherent polariton. In particular, we observed persistently long-range spin transport without dissipation. We will show our results obtained by numerically simulate time evolution of Eqs. of motion 1 and 2 in the following.

First, we obtained time evolution of average spin current $\langle J_{s_{x,x}} \rangle$ as shown in the Fig. 4, where polariton condensate is non-resonantly excited by 6 pumping lasers with linear polarization. In the early stage, a large-amplitude oscillation appears within a short time when switching on the pumping lasers. With time evolution, oscillation decays very quickly and disappear at 60 unit of time. Above phenomena are very similar to those of average density current shown in Figures 3. However, it is interesting to point out that such large-amplitude oscillation has very asymmetric behavior in contrary to symmetric behavior in average density current. Such asymmetric phenomenon may be understood from the symmetry breaking by effective magnetic field stimulated by the pumping lasers. Importantly, the remaining question is how to generate steady spin current without dissipation. Therefore, we try to deal with such question by manipulating pumping laser.

Interestingly, persistent spin current is quickly developed at time 1050 and it's amplitude is centred at 0.15. Moreover, the fast small-amplitude oscillation still appear which may be understood as stimulating surface oscillation mode by breaking symmetry on the spatial distribution of pumping lasers. Next, we compare the spin current along the different directions. Interestingly, average spin current $\langle J_{s_{x,y}} \rangle$ along the y direction has

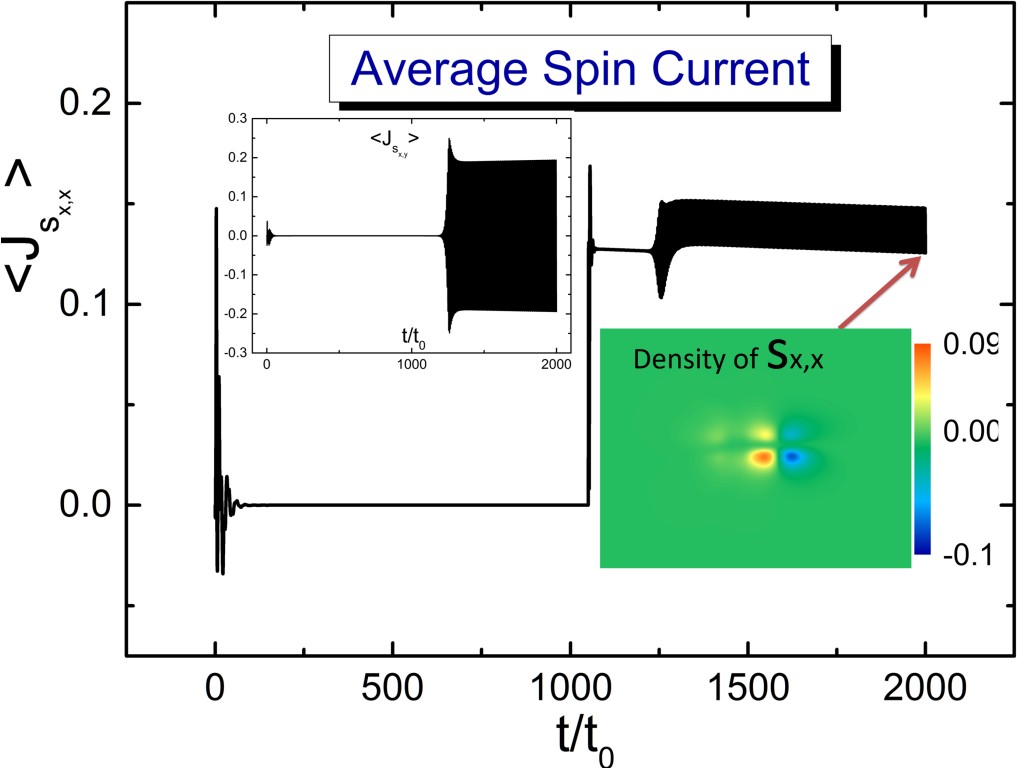

Figure 4: (Color online) Normalized average spin current $\left\langle J_{s_{x,x}} \right\rangle$ of condensate which is non-resonantly excited by 6 points laser with linear polarization. The insets shows density profile of the spin current $J_{s_{x,x}}$ at the final stage after shifting position of two middle lasers along the x direction, and that for normalized average spin current along the y direction $\left\langle J_{s_{x,y}} \right\rangle$. The size of profile is 24x24 and the other parameters used in the simulations are the same as those in the Fig. 1.

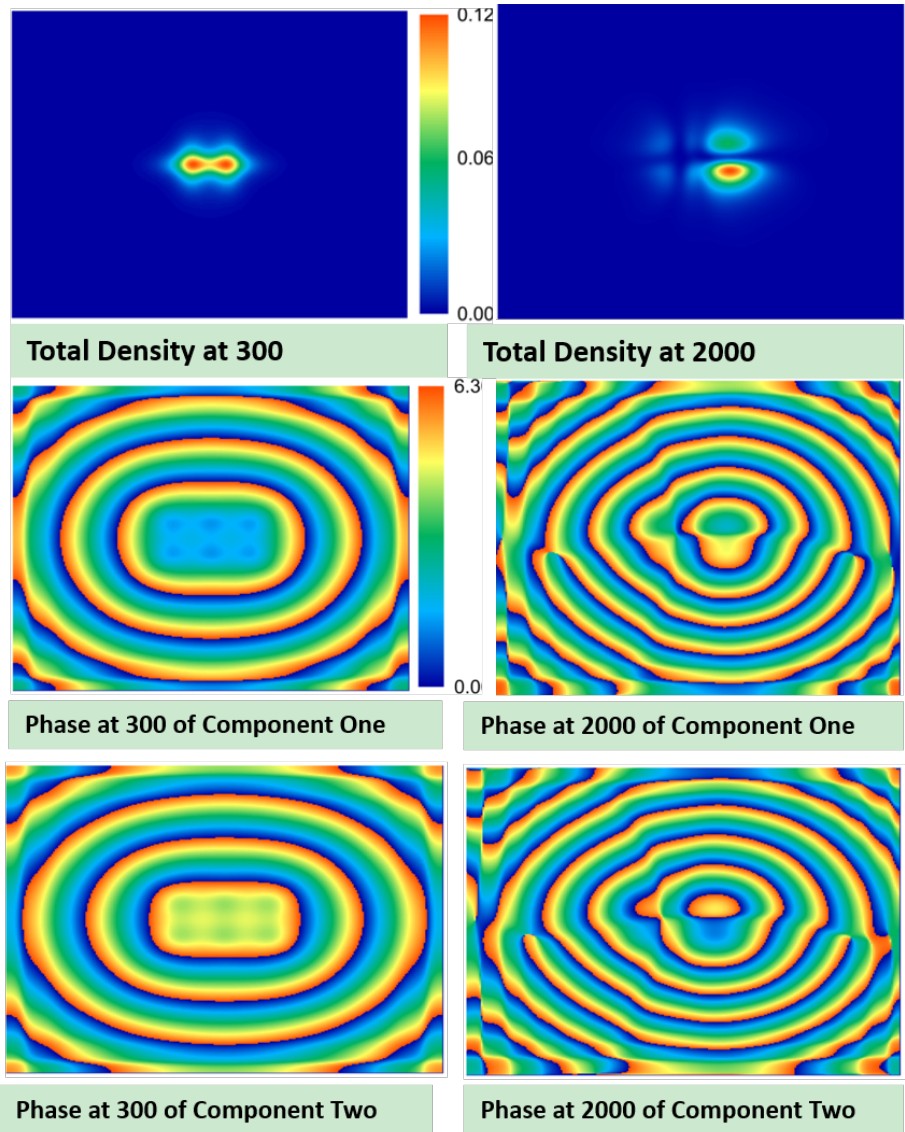

Figure 5: (Color online) Total density and each phase profile of condensed polariton excited by 6 points laser with linear polarization. The left column and right column correspond to the spatial distributions before and after shifting position of two middle lasers along the x direction, respectively. The size of profile is 24x24 and the other parameters used in the simulations are the same as those in the Fig. 1.

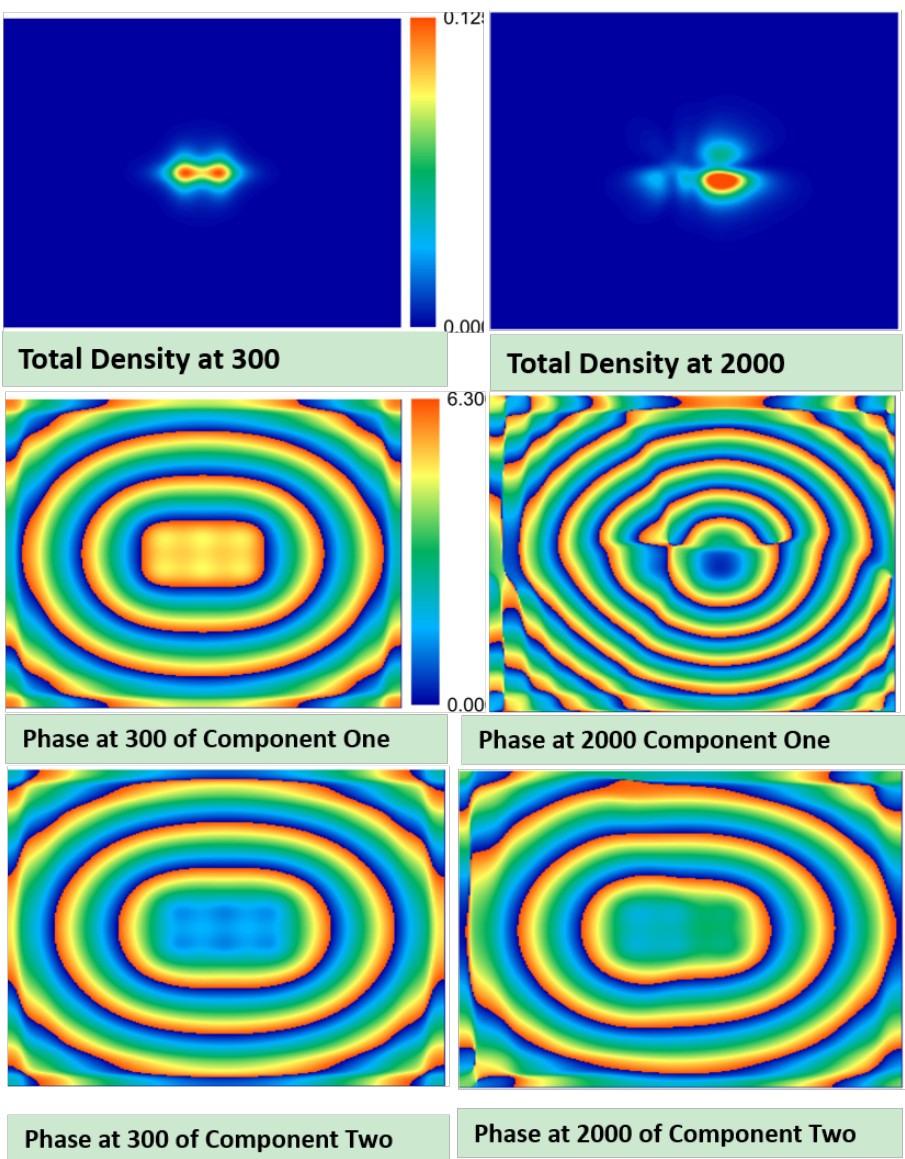

Figure 6: (Color online) Total density and each phase profile of condensed polariton excited by 6 points laser with circular polarization. The left column and right column correspond to the spatial distributions before and after shifting position of two middle lasers along the x direction, respectively. The size of profile is 24x24 and the other parameters used in the simulations are the same as those in the Fig. 1.

dramatically different behavior as shown in the inset of Fig. 4. As we can see that $\left\langle J_{s_{x,y}} \right\rangle$ represents very symmetric oscillation centred at zero value. Such different behavior between $\left\langle J_{s_{x,x}} \right\rangle$ and $\left\langle J_{s_{x,y}} \right\rangle$ is due to shift lasers' position along the x direction instead of y direction. Therefore, we can conclude that net spin current may be induced by breaking symmetric distribution of pumping lasers along preferred direction. It must be pointed out that local spin current $S_{x,x}$ may be positive or negative value as indicated in the insets of Fig. 4. Such nondissipative spin current is induced by the effective magnetic field with density- or current-dependence function. Importantly, manipulation of such effective magnetic field may be utilized to generate various polarization textures as well as spin-polarized vortices. Now, the question is how to generate stable topological defects in our studied system.

## 6.3   Phase Slip

As is well known, condensed polariton provides a very promising platform to generate and control spin current and various spin textures through manipulating effective gauge fields (like Dresselhaus and Rashba fields). In particular, there are many kinds of quantum phases in spinor quantum fluids can be accessible experimentally in this platform. For example, there may generate fascinating topological defects by manipulating pumping lasers [49, 50, 52, 54].

Physically, in order to generate topological defects, large phase fluctuations must be occurred by reducing the coherence length and amplitude of the order parameter (polariton condensate). Therefore, let us first study how to generate large phase fluctuations. In order to generate that, we suddenly moved the position of the pumping lasers in the middle site, and then see how the phase fluctuations are formed dynamically.

Figures 5 shows the total density and each phase profile of condensed polariton before and after moving the lasers in the middle site, where each component of condensed polariton is illuminated with the same laser power. Interestingly, while condensed polaritons are concentrated on the right part, large phase disturbance has been generated for each component. In particular, in low density region, there are large phase fluctuations where ring-shape phase jump shows the behavior of splitting and joining together. Physically, due to energy advantages, topological defects are initially formed in low-density regions, then, due to the dissipation of energy, these topological defects gradually moved to the high-density area and eventually reached a stable state. Therefore, we can expect that it is very promising to produce stable topological defects (such as quantum vortices) in such system.

Furthermore, we want to control which component of the condensed polaritons will generate large phase fluctuations. Therefore, each component of condensed polariton is illuminated with the different laser power and then see how the phase fluctuations are formed dynamically. Figures 6 shows the total density and each phase profile of condensed polariton before and after moving the lasers in the middle site. Interestingly, the phases of the two components have very different shape distributions. Here, large phase disturbance has been generated for component one which was illuminated with the laser power, however, there is not much change in the phase of the second part. Moreover, in component one, large phase fluctuations are closer to high-density areas where ring-shape phase jump shows the behavior of splitting and joining together.

It must be admitted that stable topological defects are not created as they require reconfiguring a large number of spins and density at a large energy cost. Generally, what kinds of stable topological defects are developed depending on the dynamics of gauge potential together with vector field, such as Maxwell-Chern-Simons-vector Higgs model for the the superconductivity of $Sr_2RuO_4$ [61]. In our studied non-equilibrium exciton-

polaritons liquid, spin- and density-dependent effective gauge fields play important roles on the phase fluctuations and make effective gauge fields more controllable comparing with conventional solid state system and ultracold atoms. Finally, we remark that the physics described in our study may be generally applicable to the recovery of complex order parameters in other systems. Photoinduced phase fluctuations may be crucial to understanding the mechanism of photoinduced superconductivity in the striped cuprates. These phenomena can be conveniently probed by real-space spectroscopy, and phase imaging [46].

# 7 Conclusions.

In conclusion, we have demonstrated a practical way to control spin polarization, generate density and spin current, and induce large phase fluctuations in an exciton-polariton condensate. For the polariton lifetime, The above-mentioned behaviors can be readily excited in photoluminescence experiments and detectable by the time-resolved micro-photoluminescence spectroscopy [49, 89] or spin noise spectroscopy [90, 91]. Our results are of particular significance for creating these excitations in experiments and for exploring novel phenomena associated with them. This noticeably spin amplification and spin transport could offer a promising way to optimize spin signals in future devices with using polariton condensates.

# Acknowledgements

We are grateful to N. Berloff for discussions.

**Funding information**  The financial support from the early development program of NanChang University and Skoltech-MIT Next Generation Program is gratefully acknowledged.

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
