# Peer review of "Dynamical Generation of Spin Current and Phase Slip in Exciton-Polariton Condensates"

_SciPost Physics_

## Round 1 · Referee Report · Anonymous (Referee 1) · 2022-9-4

Report

Dear Editor,

In manuscript scipost_202104_00013v1, the authors describe and simulate spin and phase of cavity exciton-polaritons with different excitation configurations. The manuscript has interesting contents but does not meet the criteria for publication in SciPost Physics.

Expectations (at least one required) - the paper must: Detail a groundbreaking theoretical/experimental/computational discovery; No. Present a breakthrough on a previously-identified and long-standing research stumbling block; No. Open a new pathway in an existing or a new research direction, with clear potential for multipronged follow-up work; Possibly. Provide a novel and synergetic link between different research areas. No.

General acceptance criteria (all required) - the paper must: Be written in a clear and intelligible way, free of unnecessary jargon, ambiguities and misrepresentations; No. Contain a detailed abstract and introduction explaining the context of the problem and objectively summarizing the achievements; Yes. Provide sufficient details (inside the bulk sections or in appendices) so that arguments and derivations can be reproduced by qualified experts; Almost all details are present. Provide citations to relevant literature in a way that is as representative and complete as possible; Yes. Provide (directly in appendices, or via links to external repositories) all reproducibility-enabling resources: explicit details of experimental protocols, datasets and processing methods, processed data and code snippets used to produce figures, etc.; Code not provided. Contain a clear conclusion summarizing the results (with objective statements on their reach and limitations) and offering perspectives for future work. Yes.

The strengths of this manuscript include: 1. A detailed mathematical description of spin physics in cavity exciton-polaritons. 2. Some clues towards experimentally realizing spin currents. 3. There are extensive references.

The main weakness of the manuscript are: 4. The primary use of the work is that the calculations might be repeated to match experimental conditions, but the authors did not provide any code for their simulations. This is a requirement for SciPost. More computational details might also add insight into the precision of the calculations. 5. I am not convinced that the sudden changes in pump geometry that were simulated are experimentally realizable. From an experimental perspective, relocating a laser beam within 6 ps without any side-effects does not sound right.
6. Several figures are poorly designed, to the point that they are uninterpretable.

Some more detailed comments that I have are: 7. All figures are missing axis labels and/or unit labels, including on color scales. 8. All figures: "Profile" is unclear. 9. The manuscript refers to polarization one/two, instead of using conventional terms like horizontal/vertical, left/right, or s/p. 10. p. 3 "dichroism scattering" is unclear. 11. Fig. 1 "Liner" -> "Linear" 12. Fig. 1 "small radius" is unclear. Perhaps "laser spot radius?" 13. p. 5 "phenomenological coefficients to be determined experimentally." How? Are there any known constraints? 14. p. 5 "cross spin" is unclear. 15. p. 5 "is assumed nonpolarized (corresponding to linear or horizontal polarization)" is unclear. 16. p. 5 "The spatial diffusion rate of reservoir density has been neglected." It is not obvious that this is a reasonable assumption on the picosecond scale. 17. Phonon interactions are not discussed. It is not obvious to me that they are negligible. 18. Sec. 4 Usually, a condensate is a confined system. It is not clear to me that a spatially homogeneous system is relevant to condensates. 19. p. 5 "Below the pumping threshold, the condensate remains unpopulated," "Condensate" refers to occupation of the lowest quantum state. When there is no condensate, there should still be a nonzero number density occupying that quantum state. It just is not condensed. The word "unpopulated" does not seem right. 20. p. 7 Is there any reason to think g_2 and h_2 should be the same or different? 21. p. 7 "that energy gap is polarization dependence" is unclear. 22. p. 8 I could not find a definition for \lambda. 23. p. 8 "scaling units" is unclear. Perhaps "scaled units?" I suppose these are the units that are missing from the figures. 24. p. 8 "stationary solution" is unclear. Perhaps "the stationary state" or "a stationary fluid?" 25. p. 9 "energy gap is polarization dependence." is unclear. 26. p. 9 "reaching maximal value with the condensate density decreased and spin polarized away from the pumping center." is unclear. 27. p. 9 "blueshift" How is this blueshift expressed in the theory? 28. p. 9 "polarization may lower the frequency obviously under the laser power is large enough" is unclear. 29. p. 10 The offset of the center of the Gaussian profile should be motivated. 30. p. 10 "due to their interactions converted into kinetic energy of coherent polariton." is unclear. Is there more to this than potential energy converted to kinetic energy? 31. p. 10 "as shown in the Fig. 1(c), density profiles of incoherent hot exciton and polariton condensate represent linear and circular polarization" There are no labels for "density", "linear," or "circular." It is not possible to tell if the correct figure is described. 32. p. 10 What is the purpose of multiple pumping points? There are simpler ways to change pumping geometry. Why were these chosen? 33. p. 10 "the neighbouring condensed polaritons are polarized with opposite polarization as can be seen from sz distribution clearly" is unclear. 34. p. 10 "various correlation function" It is unclear what correlation functions are being discussed or their relevance. 35. p. 10 "if shifted the pumping laser is linear or circular polarized" is unclear. 36. Fig. 3 is disorganized. 37. p. 12 "The appearance of such oscillation can be understood from the large overlap of incoherent hot exciton and coherent polariton which leads to the large repulsive force in the beginning" is unclear. 38. p. 12 Why is the current described as "steady" when the laser is "suddenly" shifted? How is "steady" backed up by simulation results? 39. p. 12 "inset of Fig. 3" Which one?
40. p. 12 "as stimulating surface oscillation mode by breaking symmetry on the spatial distribution of pumping lasers." Is there more to this than "We moved a laser beam and that excited a mode?" 50. p. 16 "middle site" is unclear. 60. p. 16 "Therefore, each component of condensed polariton is illuminated with the different laser power" is unclear. 69. p. 16 "practical way." The technical requirements have not been clearly stated.

---

## Round 1 · Referee Report · Anonymous (Referee 2) · 2022-10-10

Strengths

The manuscript generalizes the theoretical results of Ref.86 on exciton-polariton condensates coupled to a reservoir in order to account for additional spin components.

Weaknesses

  • The manuscript is very hard to read. It seems more a draft rather than an accomplished work. The numerical results are presented without a clear physical discussion.

  • Some of the figures are illisible, especially Fig.3 and Fig.4. In some other figures, like Fig.5, the x and y scales are missing (no ticks).

  • The generalized theoretical model contains a plethora of additional parameters (coupling constants) , whose value are given in the bare text. It should be better put in a separate table for better visibility.

  • The concept of phase slip should be explained and the authors should tell how they recognize such topological defects in the density and phase profiles.

  • The English should be improved.

Report

The manuscript does not meet the standard of scientific writing. I cannot recommend it for publication in this journal or elsewhere.
  • validity: low
  • significance: low
  • originality: ok
  • clarity: poor
  • formatting: below threshold
  • grammar: below threshold

---

## Editorial Decision

awaiting_resubmission